# Bidirectional Six-Pack SiC Boost–Buck Converter Using Droop Control in DC Nano-Grid

**DOI:** 10.3390/s23218777

**Published:** 2023-10-27

**Authors:** Yeonwoo Kim, Sewan Choi

**Affiliations:** 1Department of Automotive Parts, Busan Machinery Research Center, Korea Institute of Machinery & Materials, Busan 46744, Republic of Korea; woo@kimm.re.kr; 2Department of Electrical & Information Engineering, Seoul National University of Science and Technology, Seoul 01811, Republic of Korea

**Keywords:** DC nano-grid, droop control, bidirectional boost–buck converter, six-pack SiC-IPM, ESS

## Abstract

This paper proposes a bidirectional boost–buck converter employing a six-pack SiC intelligent power module using droop control in DC nano-grids. The topology is constructed as a cascaded structure of an interleaved boost converter and buck converter. A six-pack SiC intelligent power module (IPM), which is suitable for the proposed cascaded structure, is adopted for high efficiency and compactness. A hybrid control scheme, in which holding a particular switch always results in a turn-off or turn-on state according to the boost mode and the buck mode, is employed to reduce the switching losses. By applying the hybrid control scheme, the number of switching operations of the switches can be minimized. Since switchover of the current controller is not required, smooth transition is enabled not only from the buck mode to the boost mode but also vice versa. As a parallel control, a secondary control is employed with DC droop control, which has a trade-off relationship between voltage sag and current sharing. It is possible to enhance the accuracy of current sharing while effectively regulating the DC link voltage without voltage sag. This is verified experimentally using two modules as laboratory prototypes, of which the power rating is 20 kW each.

## 1. Introduction

With the increasing demand for distributed power generation systems and energy storage systems (ESS) in remote islands and mountainous areas, research on small-scale power grids has been actively underway [1,2,3,4,5,6,7,8,9,10]. Nano-grids also fall into the category of small-scale power grids; their rated power is 1 MW or less, which is lower than that of micro-grids [11,12]. DC nano-grids do not experience issues related to stability, frequency, synchronization, and reactive power, unlike AC nano-grids. DC nano-grids are also advantageous in that they allow DC power generation systems, such as solar photovoltaic (PV) systems and fuel cells, to feed DC loads without secondary power conversion [13,14]. Figure 1 presents the overall structure of a DC nano-grid system composed of ESS, PV systems, uninterruptible power supplies (UPS), electric vehicles (EV), and DC loads. In general, when the grid voltage is three-phase 380 V, the DC link voltage ranges from 750 V to 800 V. DC–DC converters are designed to cover a wide range of voltages in order to allow the use of various types of batteries, which can be employed in ESS or EV. Moreover, bidirectional converters have a very important role in DC nano-grids. The representative topology of a bidirectional converter is DAB, and research on bidirectional converters, such as the various control methods of DAB and variants of DAB hardware, is being actively conducted [15,16,17,18,19,20]. However, in order to apply the various batteries used in ESS and EV to DC nano-grids, a buck–boost-type topology is needed, rather than a buck-type or boost-type topology. This is due to it being used to cope with cases where the battery voltage is higher or lower than the DC link voltage when the DC link voltage is fixed. Furthermore, it is worth noting that DC–DC converters are required to be able to step up and step down the voltage during both the charging and discharging of batteries [21,22]. Converters, which allow for bidirectional power flow and are able to step up and step -down the voltage while charging and discharging, include the synchronous rectification buck–boost converter, the Ćuk converter, and the SEPIC converter. However, these converters have a disadvantage; the voltage rating of their switch corresponds to the sum of input and output voltages, thereby it is very high [23,24,25,26,27,28,29].

In contrast, a cascaded converter is not only capable of bidirectional operations and stepping up and stepping down the voltage during charging and discharging but the voltage rating of its switch is also low. Therefore, cascade-type converters for voltage stepping up/down are suitable for applying the various batteries used in ESS or EV to the DC nano-grid. Cascaded converters can be classified into cascaded buck–boost converters and cascaded boost–buck converters according to the combination order. In general, an interleaving method is applied to decrease the current rating of the switch, which makes switch selection easier and reduces the volume of passive devices. The conventional cascaded buck–boost converter is interleaved, and its topology is presented in Figure 2a. In general, modular switching devices are preferred to obtain converters with a higher efficiency and a more compact structure. Thus, six-pack intelligent power modules (IPM) are often employed. However, IPM cannot be used in an optimum manner in a cascaded buck–boost converter, because the number of MOSFETs are mismatched. The converter requires both the input and output stages to have the same number of phases for interleaving. Even though the interleaving cascaded buck–boost converter transforms asymmetrically, as shown in Figure 2b,c, the IPM is not used optimally, since high-side MOSFET (*Q*_1_, *Q*_3_, *Q*_5_) drains are disconnected partially. In Figure 2b, the drains of *Q*_1_ and *Q*_3_ are disconnected, and the drains of *Q*_3_ and *Q*_5_ are disconnected in Figure 2c. In order to use the six-pack IPM optimally, high-side MOSFET drains should be connected to each other.

Meanwhile, in DC nano-grids, converters are often modularized for various reasons, for example, better capacity scalability, easier system maintenance, and improved reliability [30,31]. Modular converters require parallel operation control methods for accurate current sharing. Among the parallel operation techniques, droop control is advantageous in that it requires no communication between modules to achieve load sharing and can be installed wherever necessary regardless of site conditions [32,33]. However, load sharing between modules may be inaccurately conducted due to differences in the line impedances connecting the converter to the load, thereby leading to voltage sag [34,35,36].

This paper proposes a high-efficiency bidirectional modular converter with a wide voltage range capable of the stepping up and stepping down the voltage. The converter modules are connected in parallel for current sharing, and each module is constructed to have a cascaded structure comprising a two-phase interleaved boost converter and a single-phase buck converter. In order to make the converter higher in efficiency and more compact in structure, modular devices are preferred. Among them, a six-pack SiC-based IPM is optimally employed through the proposed topology configuration. In addition, a hybrid switching control scheme, in which a particular switch always held in a turn-off or turn-on state according to the boost mode and buck mode is applied to minimize the number of switching operations, thereby reducing switching losses. Each converter module is controlled using an algorithm, for which the switchover of the current controller is not required. This enables the smooth transfer from the buck mode to the boost mode and vice versa. Furthermore, the application of parallel control, combined with the DC droop control and secondary control algorithms, is found to increase the accuracy of current sharing while enhancing the ability of the system to compensate for voltage sag.

Section 2 and Section 3 of this paper are described as the structure and configuration of the proposed DC nano-grid system and the bidirectional converter, respectively. In Section 4, a description of parallel operation control is provided. The experimental results are presented and discussed in Section 5, and the performance of two 20 kW prototypes are experimentally verified.

## 2. DC Nano-Grid System

The design specifications of the DC nano-grid system proposed in this paper are summarized in Table 1.

Figure 3 illustrates the power flow of the overall system, which is composed of ESS, PV generators, grid, and loads. For the current sharing of the 80 kW ESS, four 20 kW bidirectional converters are connected in parallel. Figure 3a shows the case where the grid is connected properly. In this case, the AC–DC converter controls the DC link voltage. When the amount of load exceeds the amount of power generated by the PV system, the DC–DC converter of the ESS feeds the load through constant power (CP) control. In contrast, when the amount of load is smaller than the amount of power generated by the PV system, the DC–DC converter charges the battery through constant current–constant voltage (CC–CV) control. Figure 3b shows the case wherein the electrical power goes out unexpectedly. This causes the grid to be shut down. In this case, the converter of the ESS controls the DC link voltage. In order to increase the performance of current sharing between the converter modules while improving the system’s ability to compensate for the DC link voltage sag, DC droop control and secondary control algorithms are applied. When the amount of load is larger than the amount of PV power generation, the converters discharge the battery, and when the amount of load is smaller than the amount of PV power generation, the converters then charge the battery.

## 3. Proposed Bidirectional Converter

### 3.1. Topology Selection

The DC link voltage is 750 V, while the battery voltage ranges from 225 to 830 V; hence, the DC–DC converter is required to not only allow for bidirectional power flow but also to buck and boost the voltage. Among converters capable of stepping up/stepping down the voltage in bidirectional power flow are cascaded converters. Cascaded converters can be classified into cascaded buck–boost converters and cascaded boost–buck converters depending on the combination order. The cascaded buck–boost converter has a cascaded structure of a buck converter in the first stage and a boost converter in the second stage. In contrast, the cascaded boost–buck converter has a cascaded structure of a boost converter in the first stage and a buck converter in the second stage [37,38,39]. Among them, the cascaded buck–boost converter additionally requires a filter due to its larger battery current ripple [40,41,42]. Moreover, when the cascaded buck–boost converter is interleaved and employs a six-pack IPM, the IPM cannot be used in an optimum manner. This is because the number of MOSFETs is mismatched. The converter requires both the input and output stages to have the same number of phases for interleaving. Even though the interleaving cascaded buck–boost converter transforms asymmetrically, the IPM is not used optimally. This is because high-side drains are disconnected partially. In order to use the six-pack IPM optimally, high-side MOSFET drains should be connected to each other. In contrast, the cascaded boost–buck converter has a smaller current ripple compared to the cascaded buck–boost converter, since input and output inductors act as filter. When interleaving is applied, the cascaded boost–buck converter, which is symmetrical in structure, cannot use the IPM optimally, because the number of MOSFETs is mismatched. However, the cascaded boost–buck converter, which is asymmetrical in structure, allows different numbers of phases for the input and output stages. Therefore, it is possible to optimally use a commercial six-pack IPM. Figure 4 shows the proposed bidirectional DC–DC converter with a cascaded structure comprising a two-phase interleaved boost converter and a single-phase buck converter.

Figure 5 shows a commercial SiC-based six-pack IPM composed of a SiC MOSFET. The IPM and its schematic diagram are presented as Figure 5a,b, respectively. The SiC MOSFET has good characteristics of switching and conducting. SiC has a lower drift layer resistance internally than Si-based switches and does not need to inject minority carriers to lower the on-resistance per unit area; high-speed switching is possible. Moreover, SiC has a higher doping concentration than Si-based switches, so it is possible to achieve low on-resistance [43,44,45]. PMF75-120-S002 (MITSUBISHI Electric Co. Ltd., Tokyo, Japan) is applied. The IPM is embedded with gate drivers and protection circuits; it also contains a relatively small number of parasitic components. The rated voltage is 1200 V, and the rated current is 75 A. The maximum current in the switch of the proposed converter is 63 A in the boost mode and 34 A in the buck mode. In other words, the converter proposed for using the IPM is suitable.

In order to compare switch losses under the same conditions, the cascaded buck–boost converter’s topology is modified asymmetrically, as shown in Figure 2c, and is compared with the proposed converter. The reason for choosing the topology as shown in Figure 2c, instead of the topology shown in Figure 2a and in Figure 2b, is that conduction loss is lowest among the three cases and the number of switches is lower than that in the symmetrical structure. Hybrid switching is known to minimize the number of switching operations according to the boost mode and buck mode, thereby reducing switching losses and achieving high efficiency. With hybrid switching applied to the topologies of Figure 2c and the proposed converter, Figure 2c and the proposed converter are compared with regard to the maximum voltage and current of the switch, parameters of inductors and capacitors, and also to the necessity of an additional filter as shown in Table 2. The total number of switches is the same, but the number of inductors and capacitors is greater in the proposed topology than in Figure 2c’s topology. In general, passive devices account for a large portion of the overall volume of the converter. Given that the proposed topology has a small battery current ripple and does noy require the addition of a filter, the proposed topology is similar to the topology shown in Figure 2c in terms of volume.

Figure 6 shows a comparison of estimated power losses resulting from the switches using Equations (1) and (2) [46,47] when hybrid switching is applied to both topologies.
(1)$$P_{switching}=[0.5I_{D}V_{DS}f_{sw}(t_{on}+t_{off})]+[0.5C_{oss}V_{DS}^{2}f_{sw}]$$
(2)$$P_{conduction}=I_{rms}^{2}R_{DS(on)}$$

The conduction losses of the two topologies are compared in Figure 6a. The proposed topology shows a smaller conduction loss due to a lower switch current. As shown in Figure 6b, there is no significant difference in the switching loss because hybrid switching is equally applied to both topologies.

### 3.2. Hybrid Switching Technique

In the proposed converter, switches in each leg operate in a complementary manner. Instead of operating all six switches each time, the proposed converter allows switches to operate through hybrid switching, as shown in Figure 7. Main duties *D*_1_ ~*D*_3_ are matched with main switches *Q*_2_*, Q*_4_*,* and *Q*_5_, respectively. With the battery as an input and the DC link as an output, in the boost mode, switches from *Q*_1_ to *Q*_4_ execute switching operations, *Q*_5_ remains turned on, and *Q*_6_ remains turned off, as shown in Figure 7a. During boost mode, the voltage of the capacitor *V*_dc_, of which the middle position of the topology, is equivalent to the DC link voltage. In the buck mode, as shown in Figure 7b, switches *Q*_5_ and *Q*_6_ perform switching operations, *Q*_1_ and *Q*_3_ remain turned on, and *Q*_2_ and *Q*_4*,*_ remain turned off. During buck mode, the voltage of capacitor *V*_dc_ is equivalent to the battery voltage. In contrast, with the DC link as an input and the battery as an output, the switching patterns are reversed. In other words, switching operation in the boost mode proceeds as shown in Figure 7b, and switching operation in the buck mode proceeds as shown in Figure 7a. This paper discusses the proposed topology and control scheme, based on batteries as input and DC links as output.

Hybrid switching is applied to the proposed bidirectional converter to minimize the number of switching operations according to the buck mode and boost mode. Thereby, switching losses are reduced. Furthermore, the converter would constitute a two-stage converter with a slow output response if all six switches operated together each time, as employed in the conventional method. Hybrid switching enables the converter to operate in a single-stage configuration. As a result, the converter’s output response is faster compared to that of the conventional method. However, a large transient may occur during the transition between buck mode and boost mode, leading to overvoltage, overcurrent, and switch failure. Thus, it is necessary to employ a control algorithm to implement a smooth transition between buck mode and boost mode.

### 3.3. Control Algorithm to Implement Seamless and Autonomous Mode Transition

Figure 8 shows a control algorithm designed to enable a seamless and autonomous mode transition from the buck mode to the boost mode and vice versa. It is composed of two voltage controllers and three current controllers, which achieved average value control using a conventional PI (Proportional–Integral) compensator. Each compensator is saturated or activated, depending on the operation mode of the proposed converter. When the saturated compensator becomes activated, a cumulative error from the saturated integrator may lead to the malfunction of the compensator. In order to prevent malfunction, anti-windup is applied. The parameters of control are listed in Table A1 of Appendix A.

When the grid works properly, the external feedback loop is connected to the battery voltage (*V*_Bat_) controller, point (C) in Figure 8. However, in the case of grid failure, the external feedback loop is connected to the DC link voltage (*V*_Link_) controller, point (F), in order to execute DC link voltage control. Moreover, the internal feedback loop conducts current control according to the reference value of battery current *I*_B_ref_ required by the external feedback loop. In order to keep each phase current of the interleaved boost converter well balanced, the inductor current of each phase (*i*_L1_ and *i*_L2_) is controlled, through using each current controller. Furthermore, the feed-forward duties *d*_buck,ff_ and *d*_boost,ff_ are added to the internal feedback loop to improve control performance and disturbance suppression performance. The feed-forward duties *d*_buck,ff_ and *d*_boost,ff_, which range from 0 to 1, are as described in Equations (3) and (4). This means that when the value of *d*_buck,ff_ or *d*_boost,ff_ exceeds 1, it is saturated to 1. Furthermore, it is saturated to 0 when the value is below 0.
(3)$$d_{boost,ff}=1-\frac{V_{Bat}}{V_{Link}}\;(0\leq d_{boost,ff}\leq 1)$$
(4)$$d_{buck,ff}=\frac{V_{Link}}{V_{Bat}}\;\;\left(0\leq d_{buck,ff}\leq 1\right)$$

The inductor current *i*_L3_ of the buck converter is controlled according to the predicted current $i_{L3}^{\wedge }$, which is obtained from the combination of Equations (3) and (4). $i_{L3}^{\wedge }$ is calculated as described in Equation (5).
(5)$$I_{L3}^{\wedge }=\frac{1-d_{boost,ff}}{d_{buck,ff}}\cdot I_{Bat}$$

In order to assist the saturation or activation of the current control compensator, saturation parameters *ε*_force,sat,buck_ and *ε*_force,sat,boost_ are used in the internal feedback loop.

The parameters *ε*_force,sat,buck_ and *ε*_force,sat,boost_ make the current controller saturated forcibly in the buck mode and boost mode, respectively. The value of parameters is varied from 0 to the value, which is sufficient to saturate the compensator. The *ε*_force,sat,buck_ is in proportion to the difference between *V_dc_* and *V*_Link_, and the *ε*_force,sat,boost_ is in proportion to the difference between *V*_dc_ and *V*_Bat_. Considering that *V*_dc_ is equal to *V*_Link_ in the boost mode and equal to *V*_Bat_ in the buck mode, the saturation parameter values are as shown in Table 3. During boost mode, *i*_L1_ and *i*_L2_ current controllers are activated because *ε*_force,sat,buck_ is 0. Moreover, the *i*_L3_ current controller is saturated due to the influence of *ε*_force,sat,boost_. As a result, *Q*_5_ remains turned on and *Q*_6_ remains turned off during boost mode, since *D*_3_ is saturated to the value 1. In contrast, the *i*_L3_ current controller is activated and *i*_L1_ and *i*_L2_ current controllers are saturated during buck mode. Likewise, *Q*_2_ and *Q*_4_ remain turned off and *Q*_1_ and *Q*_3_ remain turned on because *ε*_force,sat,boost_ is 0, and *ε*_force,sat,buck_ is sufficient to be saturated.

Figure 9 shows simulated waveforms obtained during the seamless transition between buck and boost mode. As shown in Figure 9, seamless transition between buck and boost modes is possible, because the switchover of the current controller is not required. The simulated power rating of charging or discharging the battery is 20 kW, the DC link voltage is 750 V, and the battery voltage is varied between 225 V and 830 V. The battery current, which is sum of *i*_L1_ and *i*_L2_, is controlled while the battery voltage changes. Figure 9a shows that the battery voltage *V*_Bat_ increases from 225 V to 830 V, due to charging of the battery. While *V*_Bat_ increases, mode transition occurs from the boost mode to the buck mode. Contrastively, Figure 9b shows the decrease in the battery voltage due to discharging and also shows the mode transition from the buck mode to the boost mode. In the boost mode, considering that the feed-forward *d*_buck,ff_ becomes 1 and the influence of *ε*_force,sat,boost_, *D*_3_ is saturated to 1. Thus, *Q*_5_ remains turned on and *Q*_6_ remains turned off. Meanwhile, switches from *Q*_1_ to *Q*_4_ execute switching operations since the values of *D*_1_ and *D*_2_ fall within the range of the triangle wave, which is a carrier of PWM.

In contrast, during buck mode, *Q*_5_ and *Q*_6_ execute switching operations, because *D*_3_ falls within the range of the carrier for PWM. Furthermore, given that the feed-forward *d*_boost,ff_ becomes 0 and the influence of *ε*_force,sat,buck_, *D*_1_ and *D*_2_ are saturated to 0. As a result, *Q*_1_ and *Q*_3_ remain turned on; *Q*_2_ and *Q*_4_ remain turned off. It is found that seamless and autonomous transition between buck and boost modes is achieved because the switchover of the current controller is not required regardless of the step-up or step-down situation. Figure 10 shows simulated waveforms obtained during transition from discharging to charging and vice versa. Figure 10a,b are obtained during boost mode and during buck mode, respectively. Seamless transition between charging and discharging of the battery is possible due to the original characteristic of the bidirectional topology.

## 4. Parallel Operation Using DC Droop Control

In general, the current sharing performance between modules is degraded due to a difference in line impedance during parallel operation. In order to overcome the degradation, droop control using virtual impedance is applied. Droop control is a distributed parallel operation method, and it is controlled by calculating the reference value of voltage according to the output current of each converter. Moreover, the droop control has an advantage in that it does not require a communication between modules to achieve load sharing. This is because there is a profile between the output current of each converter and the output voltage, using virtual resistance. As a result, it is possible to install it wherever necessary regardless of site conditions.

Figure 11a shows a brief configuration of the parallel connection for circuit modules to analyze droop control. According to the difference between line resistances *R*_Line1_ and *R*_Line2_ and also the difference between virtual resistances *R*_d1_ and *R*_d2_ of each converter, the output currents *i*_o1_ and *i*_o2_ of each converter are unbalanced, as shown in Figure 11b. Furthermore, if virtual resistance is much larger than line resistance, the output currents *i*_o1_ and *i*_o2_ are the same because it is possible to neglect the line resistance, in the case of equal virtual resistance. The output voltage *V*_o_ when droop control is applied can be expressed as shown in Equation (6).
(6)$$V_{o}^{*}=V_{Link}^{*}-R_{d}\cdot i_{oN}$$

Equation (6) serves as a profile in droop control. The reference value of *V*_o_ is a voltage set point *V^*^*_o_. The voltage set point is adjusted and controlled according to the output current for load current sharing. However, load sharing between converter modules may be inaccurately conducted due to differences in the line resistance. Thus, it is important to adjust the virtual resistance appropriately.

Figure 12 shows the droop slopes resulting from the difference between the line resistances and the difference between virtual resistances. Figure 12a shows case 1, for which the offset error Δ*V*_n_ is 0 and the line resistances are different from each other. In contrast, Figure 12b shows case 2, in which the offset error is not 0 and the line resistances are the same. The solid line indicates the case where the virtual resistance *R*_d_ is smaller, while the dotted line indicates the case where the virtual resistance *R*_d_*’* is larger.

The difference between *i*_o1_′ and *i*_o2_′ is smaller than the difference between *i*_o1_ and *i*_o2_. In other words, the load sharing performance is better in the case of *R*_d_’ than in the case of *R*_d_. Assuming that the virtual resistance is smaller, the effect of line resistance becomes larger. It should be noted that this leads to a degradation in load sharing performance. In contrast, if the virtual resistance is larger, load sharing performance may be improved, but the amount of the voltage sag increases, as is the case when *V*_Link_’ is lower than *V*_Link_. The trade-off relationship between load sharing performance and voltage sag is affected by a difference in virtual resistance. In order to relieve the trade-off relationship, secondary control is used with droop control. Secondary control is a method used to prevent voltage sag, which occurs when the virtual resistance is large in droop control.

Figure 13 shows the algorithm of DC droop control with secondary control. In secondary control, the central controller controls the DC link voltage, which has been reduced, in order to compensate for the DC link voltage. The central controller transmits the compensation value ∆*V*_Link_ to each converter’s controller, through low-bandwidth network communication. As a result, the value ∆*V*_Link_ is added to the voltage set point *V^*^*_o_.

Figure 14 shows the droop slope with secondary control. As in Figure 12a,b and Figure 14a shows case 1, for which the offset error Δ*V*_n_ is 0 and the line resistances are different from each other. Additionally, Figure 14b shows case 2, for which the offset error is not 0 and the line resistances are the same. The solid lines and dotted lines indicate cases before and after the application of secondary control, respectively. As a result, the compensation of the voltage sag is achieved using secondary control with the DC droop control without degradation in the load sharing performance.

## 5. Experimental Results

In order to verify the performance and feasibility of the proposed bidirectional converter intended for the DC nano-grid system, a set of two 20 kW prototype modules are manufactured and configured, as shown in Figure 15. The overall dimensions of one set are 650 mm × 445 mm × 130 mm (37.6 L). A dual-core DSP TMS320F28377D is used as a digital controller, and each module is controlled by one core.

Figure 16 shows experimental waveforms of the proposed converter during the discharging operation. To be more specific, the experimental waveform during boost mode is presented in Figure 16a, and the waveform during buck mode is shown in Figure 16b. The experimental result of Figure 16a confirms that there is no ripple component in the inductor current *i*_L3_ of the buck converter parts, and ripple components are found in the inductor currents *i*_L1_ and *i*_L2_ of the boost converter parts in the boost mode. This is because the hybrid switching control scheme is applied to the proposed converter. In contrast, Figure 16b shows that there is no ripple component in *i*_L1_ and *i*_L2_, and a ripple component is found in *i*_L3_ during buck mode. Figure 16c shows an experimental waveform of seamless mode transition between boost mode and buck mode, while the battery voltage is varied. During the experiment shown in Figure 16c, the proposed converter maintains the DC link voltage at 750 V. In this case, it should be noted that the proposed converter has seamless mode transition and controls the DC link voltage when a grid failure occurs.

Figure 17 presents waveforms experimentally obtained during droop control according to the virtual resistance *R*_d_. Figure 17a shows cases before and after the application of droop control, when *R*_d_ is 0.1 Ω. Furthermore, Figure 17b shows the case when *R*_d_ is 3.0 Ω. The more the virtual resistance is increased, the better the load sharing performance is, while the voltage sag is also increased, as shown in Figure 17a,b. A trade-off relationship between the load sharing performance and voltage sag is confirmed while droop control is applied. Furthermore, it is important to adjust the virtual resistance appropriately because the trade-off relationship is sensitive to virtual resistance. It makes parallel operation difficult for each converter module. In order to relieve the difficulty of parallel operation, secondary control is used, which prevents voltage sag.

Figure 18a,b show experimental waveforms before and after the application of secondary control, respectively. Before secondary control is applied, the voltage sag increases as the load increases. This is because the large virtual resistance improves the load sharing performance. However, after secondary control is applied, voltage sag does not occur even though the load increases, and the load sharing performance is improved.

Figure 19 shows the measured efficiency of the prototype for the proposed converter, which applies a hybrid switching control scheme. The loss of the proposed topology mainly consists of the conduction loss and switching loss of the switch. The conduction loss of the switch is greatly affected by the operating power and is determined by the current according to the battery voltage. Switching losses are determined by battery voltage and current. Thereby, the efficiency is presented according to the operating power and battery voltage. It is measured by a WT3000 (YOKOGAWA). When the efficiency is measured during the battery voltage is 225 V, the efficiency is not measured more than 12 kW. Since operating point of over the 12 kW is exceeds the electrical load equipment specifications. The measured efficiency during discharging of the battery is shown in Figure 19a. The maximum efficiency is 98.9%, while the rated efficiency is 98.8%. Figure 19b shows the measured efficiency during charging of the battery. The maximum efficiency and the rated efficiency are 99.2% and 98.8%, respectively. The experimental results confirmed that high efficiency is achieved by using the SiC-based six-pack IPM optimally and also by applying the hybrid switching control scheme for the proposed converter.

## 6. Conclusions

This paper proposes a bidirectional boost–buck converter employing a six-pack SiC optimally, using droop control with secondary control in DC nano-grid application. The topology is a cascade structure of a two-phase interleaving boost converter and a single-phase buck converter, which has a wide range of battery voltage and the capability to step up and step down in bidirectional power flow. A commercial six-pack SiC-based IPM is optimally used to implement a converter with high efficiency and compact structure. In order to minimize the switching losses, a hybrid switching control scheme, which a particular switch always hold in a turn-off or turn-on state according to the boost mode or buck mode, is applied to the proposed converter. Unlike existing converters in which all switches operate simultaneously, in aspect of the proposed converter, some switches are switching, others hold in a turn-off or turn-on state. Thus, the proposed converter has high efficiency. The maximum efficiency is 99.2%, and the rated efficiency is 98.8%. In addition, there is a smooth transition between buck mode and boost mode, because switchover of the current controller for the converter is not required. Meanwhile, as parallel operation control of the converter modules, DC droop control and secondary control are combined effectively. As a result, not only is current sharing performance improved but so too is the ability of the system to compensate for voltage sag. Experimental results are verified using two modules as laboratory prototypes, of which the power rating is 20 kW. The results from the 20 kW prototype are provided to validate the proposed topology and control schemes, which applied hybrid switching and parallel operation.

## Figures and Tables

**Figure 1 sensors-23-08777-f001:**
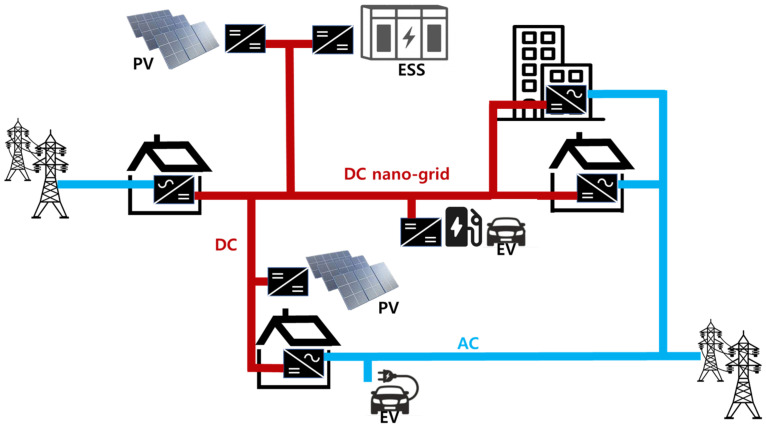
Structure of DC nano-grid system.

**Figure 2 sensors-23-08777-f002:**
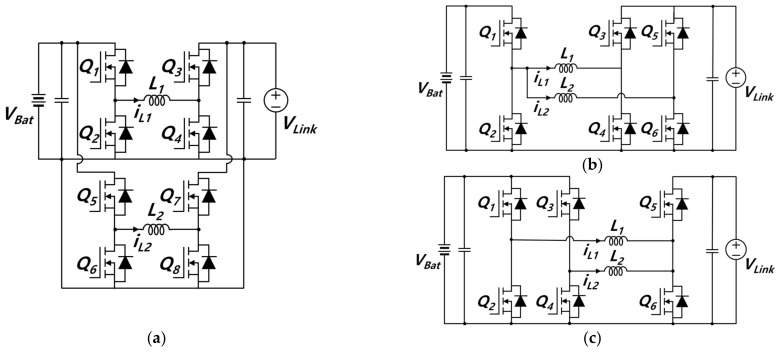
Interleaving cascaded buck–boost converter: (**a**) Symmetrical structure, (**b**,**c**) Asymmetrical structure.

**Figure 3 sensors-23-08777-f003:**
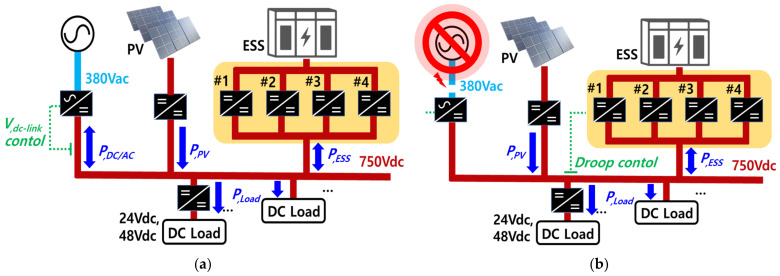
Islanding test scheme in dc nano-grid system: (**a**) Case of the grid connected, (**b**) Case of islanding.

**Figure 4 sensors-23-08777-f004:**
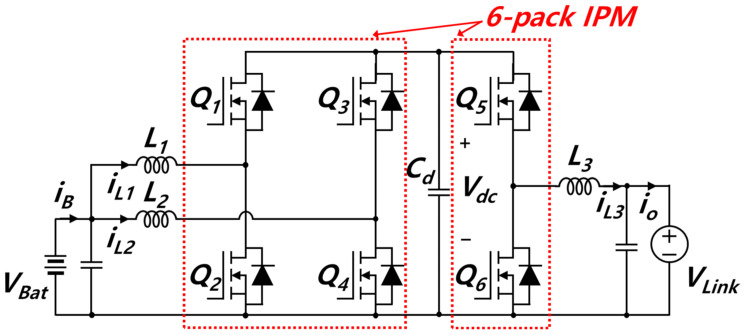
Proposed bi-directional boost-buck converter.

**Figure 5 sensors-23-08777-f005:**
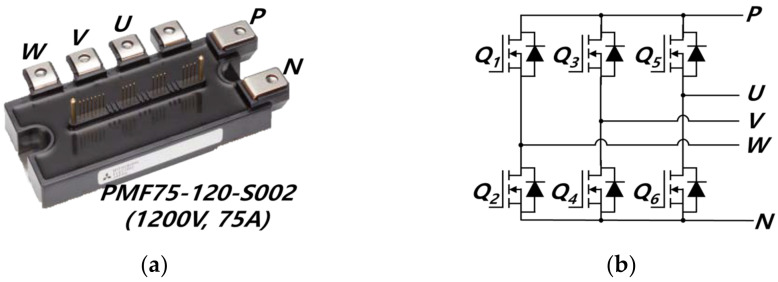
SiC-IPM: (**a**) IPM, (**b**) IPM Schematic diagram.

**Figure 6 sensors-23-08777-f006:**
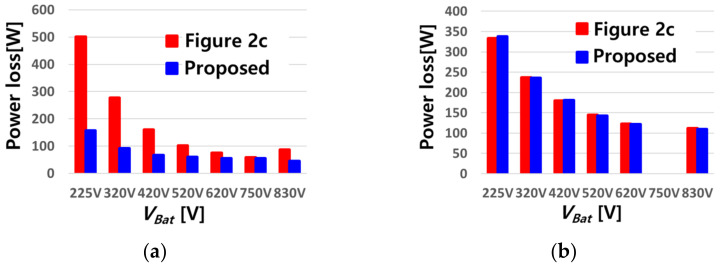
Comparison of calculated loss: (**a**) Conduction loss, (**b**) Switching loss.

**Figure 7 sensors-23-08777-f007:**
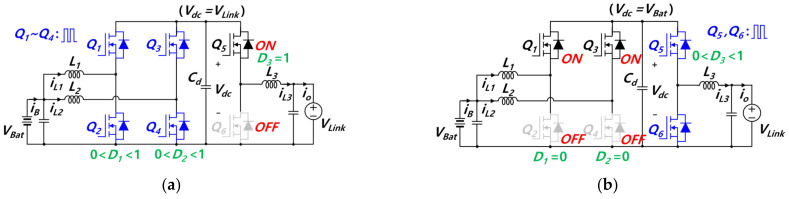
Operation mode for hybrid switching: (**a**) Boost mode, (**b**) Buck mode.

**Figure 8 sensors-23-08777-f008:**
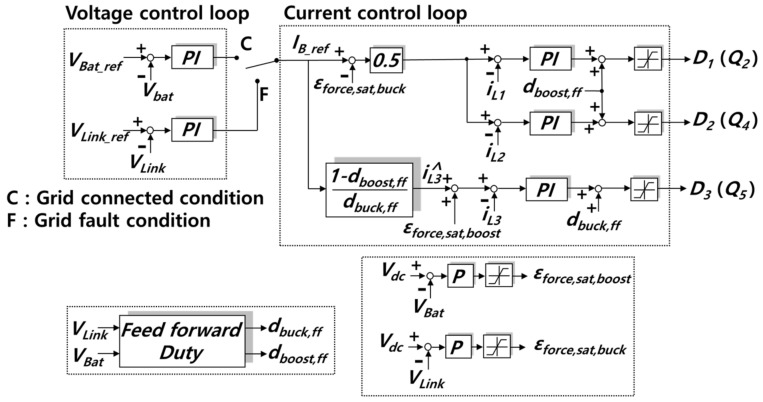
Control algorithm of the proposed converter.

**Figure 9 sensors-23-08777-f009:**
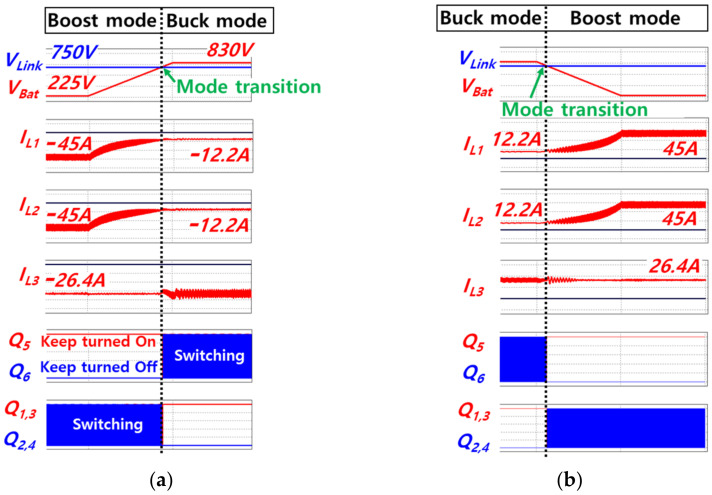
Simulated waveforms for mode transition: (**a**) During charging battery, (**b**) During discharging battery.

**Figure 10 sensors-23-08777-f010:**
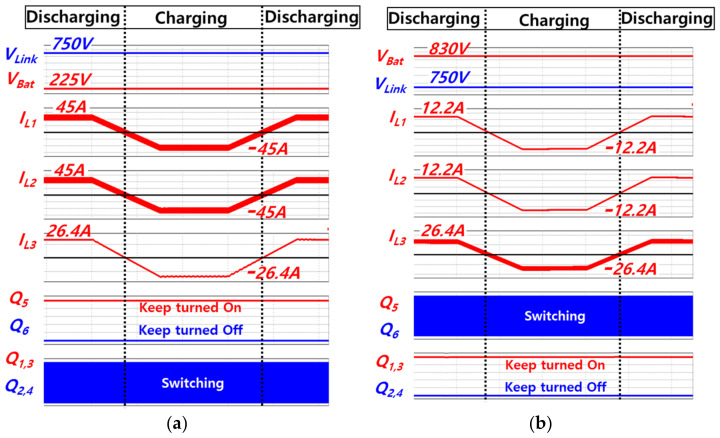
Simulated waveforms for transition from discharging to charging and vice versa: (**a**) During boost mode, (**b**) During buck mode.

**Figure 11 sensors-23-08777-f011:**
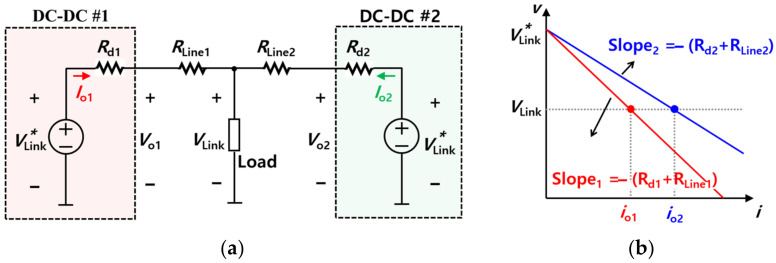
Simplified module for DC droop control: (**a**) Configuration of parallel connection for dc-dc converters, (**b**) DC droop curve.

**Figure 12 sensors-23-08777-f012:**
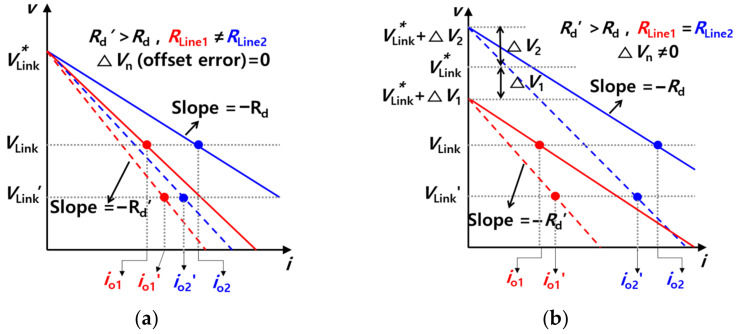
Droop slope with current sharing performance depending on slope gain (*R*_d_): (**a**) Case 1 (*R*_Line1_ ≠ *R*_Line2_, ∆*V*_n_ = 0), (**b**) Case 2 (*R*_Line1_ = *R*_Line2_, ∆*V*_n_ ≠ 0).

**Figure 13 sensors-23-08777-f013:**
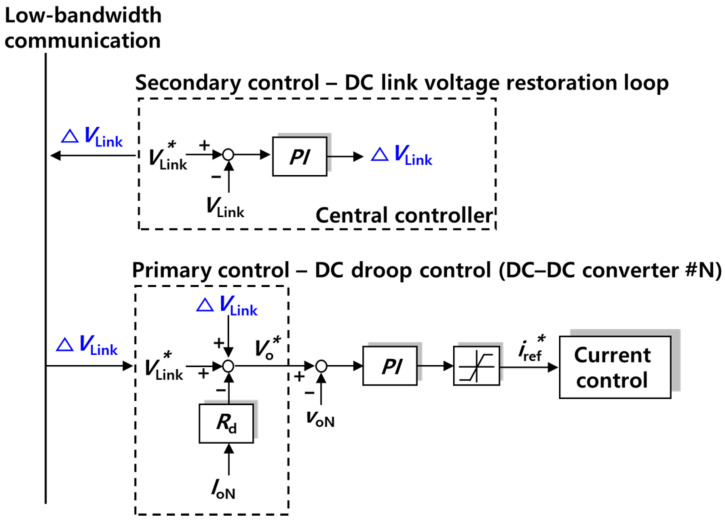
DC droop control with the secondary control algorithm.

**Figure 14 sensors-23-08777-f014:**
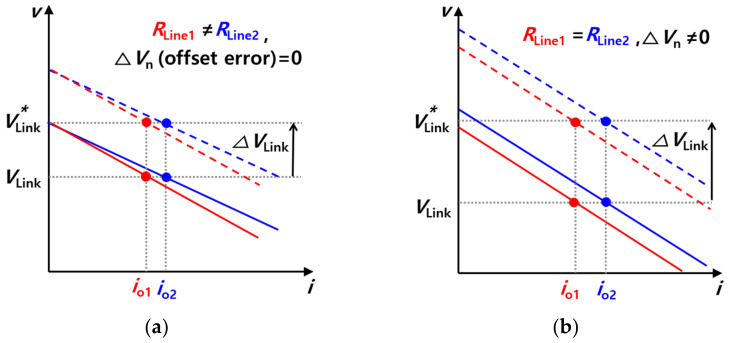
Droop slope with secondary control: (**a**) Case 1 (*R*_Line1_ ≠ *R*_Line2_, ∆*V*_n_ = 0), (**b**) Case 2 (*R*_Line1_ = *R*_Line2_, ∆*V*_n_ ≠ 0).

**Figure 15 sensors-23-08777-f015:**
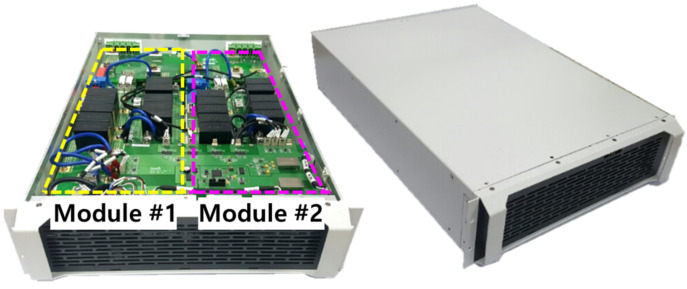
Prototype of proposed converter, which set of two 20 kW modules.

**Figure 16 sensors-23-08777-f016:**
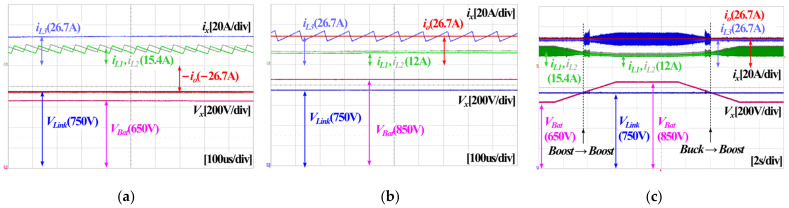
Experimental waveforms of the proposed converter in discharging operation: (**a**) *V_Link_* = 750 V, *V_Bat_* = 650 V, *P_out_* = 20 kW with boost mode, (**b**) *V_Link_* = 750 V, *V_Bat_* = 850 V, *P_out_* = 20 kW with buck mode, (**c**) *V_Link_* = 750 V, *V_Bat_* = 650~850 V, *P_out_* = 20 kW with mode transition between boost mode and buck mode, during *V_Link_* control.

**Figure 17 sensors-23-08777-f017:**
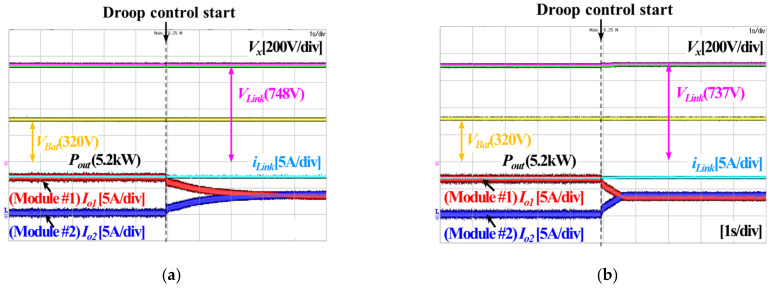
Experimental waveforms of DC droop control according to virtual resistance (*R_d_*): (**a**) *R_d_* = 0.1 Ω, (**b**) *R_d_* = 3.0 Ω.

**Figure 18 sensors-23-08777-f018:**
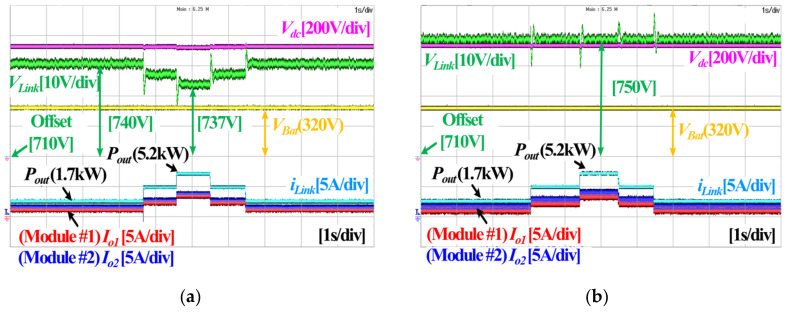
Experimental waveforms of droop control according to application of secondary control: (**a**) Droop control without secondary control, (**b**) Droop control with secondary control.

**Figure 19 sensors-23-08777-f019:**
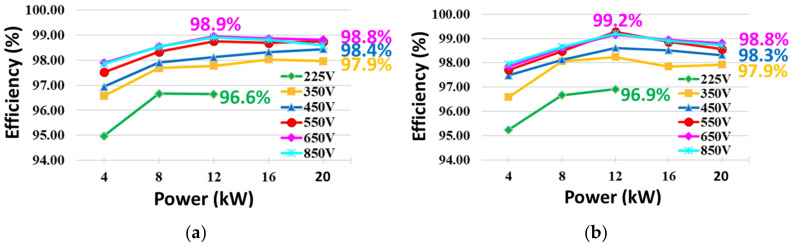
Measured efficiency: (**a**) Efficiency of discharge operation, (**b**) Efficiency of charge operation.

**Table 1 sensors-23-08777-t001:** Specification of the DC nano-grid system.

	Parameter	Value
PV	Voltage range	225–830 V
	Input power	120 kW
Battery	Voltage range	225–830 V
	Input power	80 kW
DC-DC Converter	Output power	20 kW
	DC-link voltage	700–750 V
	Nominal DC-link voltage	750 V
	Efficiency	98.5% (@20 kW)
	Charging method	CC, CP, and CV

**Table 2 sensors-23-08777-t002:** Comparison of topology.

Parameter		Figure 2c	Proposed
Switch	*V*_peak_, *I*_peak_ (No. of switches)	830 V, 63 A (2)	750 V, 63 A (2)
		830 V, 20 A (2)	830 V, 63 A (2)
		750 V, 63 A (2)	830 V, 34 A (2)
	Total number	6	6
Capacitor	*I*_rms_ (No. of switches)	27 uF, 27 A, (1)	27 uF, 2.2 A, (2)
		125 uF, 2.2 A, (1)	125 uF, 27 A, (1)
Inductor	*I*_rms_ (No. of switches)	600 uH, 56 A, (2)	600 uH, 56 A, (2)
			600 uH, 33 A, (1)
Necessity of additional filter on input and output		Yes	No

**Table 3 sensors-23-08777-t003:** Saturation parameter value according to boost mode and buck mode.

Saturation Parameter	Boost Mode	Buck Mode
*ε_force,sat,buck_*	0	α ^1^
*ε_force,sat,boost_*	α ^1^	0

^1^ Sufficient value to saturate the compensator.

## Data Availability

Not applicable.

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
