# Peer review of "Bidirectional Six-Pack SiC Boost–Buck Converter Using Droop Control in DC Nano-Grid"

_sensors, 2023, doi:10.3390/s23218777_

Round 1

Reviewer 1 Report

This paper proposes a bi-directional boost-buck converter employed 6-pack SiC using droop control in DC nano-grids. It is interesting and has practical significance. 

1) The figure 3 part should be placed in the introduction part.

2) The simulated waveforms from charging battery to discharging battery, and from discharging battery to charging battery should be added.

no

Reviewer 2 Report

Please revise the paper reflecting following comments.

1. Please add analysis why the efficiency is dependent to voltage and power rating in the experimental results presented in Figure 18.

2. Please add units of resistance in Figure 16 and the manuscript.

3. Please write or correct list of papers in the reference in uniform format. 

No specific comment about quality of English language.

Reviewer 3 Report

This paper proposed a Bi-directional Boost-Buck Converter with Droop Control. However, the reviewer has some questions.

1.     Please talk more for the background of the importance of the bidirectional converter for DC microgrid. E.g., the application of bidirectional multi-port dcdc converter for DC microgrid. Paper “Adaptive Bidirectional Droop Control for Electric Vehicles Parking Lot with Vehicle-to-Grid Service in Microgrid” could be a good reference.

2.     For the review of the bidirectional DC-DC converter, dual active bridge based is also a typical topology. Please demonstrate why dual active bridge is not selected as your topology. Paper “Model Predictive control of dual-active-bridge based fast battery charger for plug-in hybrid electric vehicle in the future grid” can be a good reference.

3.     For the proposed topology, the similar topology has already existed. Could you please justify the novelty of the proposed topology?

4.     With the proposed topology, the different switch needs to stand for different voltage stress or power stress, which is important to the reliable operation of the power converter. Could you please demonstrate more about the voltage or power stress analysis?

5.     Author claims that the maximum efficiency is 99.2%, however, without the soft switch technology, how does it realize?

6.     Could you please have a table to list all the parameters of the prototype?

not bad

Round 2

Reviewer 1 Report

2.The simulated waveforms from charging battery to discharging battery, and from discharging battery to charging battery should be added.

And the author has not given active responds in the paper. 

Author Response

I appreciate for your review about this paper.

Reviewer 2 Report

Authors revised the manuscript properly by following reviewer's comments.

Author Response

(The authors gave the same response as above.)

Reviewer 3 Report

good to publish

not bad

Author Response

(The authors gave the same response as above.)
